# Renal function impairment in cervical cancer patients treated with cisplatin-based chemoradiation: A review of medical records in a Zimbabwean outpatient department

**Pinky M. C. Manyau** [1] *, **Mensil Mabeka**[1], **Tinashe Mudzviti**[1,2], **Webster Kadzatsa**[3,4], **Albert Nyamhunga**[3,4]

1 School of Pharmacy, University of Zimbabwe, Mount Pleasant, Harare, Zimbabwe, 2 Newlands Clinic, Highlands, Harare, Zimbabwe, 3 Parirenyatwa Group of Hospitals, Radiotherapy Centre, Harare, Zimbabwe, 4 Radiology Department, University of Zimbabwe, Mount Pleasant, Harare, Zimbabwe

* pinky.manyau@gmail.com

## Abstract

### Background

There is a potential increase in risk of renal function impairment among patients with invasive cervical cancer (ICC) who are HIV-positive and treated with cisplatin-based concurrent chemoradiation (CCRT). This concern is due to overlapping nephrotoxicity of the drugs, and nephropathy from the diseases themselves. There is limited literature available for the short-term renal outcomes for HIV-positive patients with ICC during routine clinical management. This study aimed to assess if HIV-infection increased the risk of renal impairment in ICC patients treated with CCRT, and explore the respective risk factors.

### Materials and methods

This was a retrospective review of records of ICC patients treated with at least one cycle of weekly cisplatin during CCRT at the Parirenyatwa Radiotherapy Center from January 2017-December 2018. The RIFLE criteria were used to classify renal impairment. Analyses were performed with Fisher's Exact tests, Wilcoxon rank sum tests. Odds ratios (OR) were generated using logistic regression. All statistical tests were 2-sided at a 5% level of significance.

### Results

Seventy-two eligible patients were identified, 32 (44.44%) were HIV-positive. HIV-positive patients were younger (p = 0.002), had lower albumin levels (p = 0.014) and received lower cisplatin doses (p = 0.044). The mean percent reduction in estimated glomerular filtration rate (eGFR) from baseline was -19% (95% CI: -25.9% to -13.2%) for all patients. Thirty-one (43.1%) patients experienced renal impairment, 50% and 37.5% of HIV-positive and -negative patients respectively (p = 0.287). HIV-infection was associated with an adjusted OR of 1.16 (95% CI 0.35–3.43, p = 0.769). Baseline eGFR< 60ml/min was the only independent

**Data Availability Statement:** All relevant data are within the manuscript and its Supporting information files.

**Funding:** The author(s) received no specific funding for this work.

**Competing interests:** The authors have declared that no competing interests exist.

predictor of renal impairment, OR 0.25 (95% CI: 0.07–0.85). Baseline eGFR<60ml/min was also associated with receipt of lower cisplatin doses (p = 0.044).

## Conclusion

HIV-infection was not associated with elevated risk of renal impairment. Patients with an eGFR<60ml/min appear to be managed more cautiously reducing their risk for renal impairment during cisplatin therapy. The high prevalence of renal impairment in this population suggests the need for optimization of pre-treatment protocols.

## Introduction

### Background

Since the '90s invasive cervical cancer (ICC) has been classified as an acquired immunodeficiency syndrome (AIDS)-defining malignancy (ADM) [1]. The human immunodeficiency virus (HIV) prevalence in Zimbabwean ICC patients is approximately 42% [2]. The risk of invasive carcinoma in HIV-positive women has been reported to be 2 to 4 fold compared to that of their HIV-negative counterparts [3, 4]. In 2018, ICC was the leading cause of cancer related deaths among Southern African females [5]. Due to overlapping toxicities of drug therapies. Additionally, comorbid HIV-infection and ICC raises concern over a potential increase in the nephrotoxicity, and possibly chronic kidney disease (CKD).

HIV-associated nephropathy (HIVAN) is a common cause of end-stage renal disease (ESRD) in HIV-positive patients of African descent, however it does not appear to increase the risk of acute kidney injury (AKI) [6]. Up to 80% of women with ICC in Zimbabwe present with advanced disease [7]. Treatment of this patient population includes definitive cisplatin-based concurrent chemoradiation (CCRT) [8].

Cisplatin is a known cause of AKI, and prevalence of cisplatin-induced nephrotoxicity (CIN) in cervical cancer ranges from 18–35% [9, 10]. Risk factors for CIN include age ≥50 years, higher baseline estimated glomerular filtration rate (eGFR), hypoalbuminaemia, hypertension, locally advanced disease with hydronephrosis, cumulative cisplatin dose and lack of prehydration with magnesium [9, 11, 12]. However, there have been inconsistencies in risk factors identified by different studies. This is likely due to varying clinical practices in hydration protocols [13]. Additionally, premedication and dose adjustment criteria especially in the elderly, and those with renal insufficiency may also vary.

Management of HIV-positive patients with ICC may be further complicated by overlapping toxicities of tenofovir disoproxil fumarate (TDF) and cisplatin. TDF containing regimens are widely recommended for the treatment of HIV-positive adults [14], and they are the most commonly used in Africa [15]. Several case series have reported AKI with TDF [15, 16]. Albeit, the risk of TDF-induced AKI appears to be low [17, 18]. Feasibility of weekly administration of cisplatin at a dose of 40mg/m$^2$ for HIV-positive patients on combination antiretroviral therapy (cART) has been demonstrated by a phase II clinical trial [19], however there is limited data available for renal outcomes of this patient population during routine clinical management. This study aimed to assess if HIV-positive ICC patients treated with definitive CCRT are at higher risk of renal impairment, and explore the respective risk factors in Zimbabwean ICC patients.

## Materials and methods

### Study design and population

A retrospective review of medical records of patients managed with CCRT for histologically confirmed locally advanced cervical cancer (LACC) from 1 January 2017 to 31 December 2018 at Parirenyatwa Radiotherapy Center (PGH-RTC) was performed. All consecutive patients who were 18 years or older at ICC diagnosis and received at least one cycle of weekly cisplatin were included in the study.

### Data

The following data were collected for all patients: demographics, HIV-status, FIGO stage, cisplatin BSA-dose, number of weekly cisplatin cycles received, baseline and pre-treatment eGFR, cART regimen, duration of cART, albumin, body mass index (BMI), body surface area (BSA), presence of hypertension and/ or diabetes mellitus and number of cisplatin cycles. Unrecorded eGFR were calculated from the recorded serum creatinine, weight and age using the Cockcroft-Gault equation shown below.

$$eGFR = \left[ \frac{(140 - age)x\ body\ weight}{Serum\ creatinine\ \mu mol/L} \right] x\ 0.85$$

### Data analysis

Renal impairment was defined using the Risk, Injury, Failure, Loss of kidney function, and End-stage kidney disease (RIFLE) criteria. The difference of the lowest recorded eGFR from baseline was calculated. No renal impairment was defined as a <25% decrease in eGFR from baseline, Class Risk was a decrease in eGFR of 25–49% and injury was ≥50% decrease in eGFR from baseline [20].

Data were analyzed with Stata® version 13.0. Descriptive statistics were presented using medians and interquartile ranges (IQR) for continuous variables, and proportions for categorical variables. Baseline differences between HIV-positive and HIV-negative patients were tested using 2-sided Fisher's exact statistics for categorical variables and Wilcoxon rank sum tests for continuous variables. Factors associated with kidney injury were identified using univariate and multivariate logistic regression. Independent predictors of renal injury were identified using stepwise regression. The results for regression analysis were expressed as odds ratios (OR), with their respective 95% confidence interval (CI). All statistical tests were performed at a 5% level of significance.

### Ethical approvals

All data were fully anonymized before they were accessed. Ethics approvals were obtained from the Joint Research and Ethics Committee for University of Zimbabwe College of Health Sciences and Parirenyatwa Group of Hospitals (JREC#:315/19) before commencement of the study.

## Results

A total of 72 eligible records were identified, 32 (44.4%) were HIV-positive. HIV-positive patients were younger (p = 0.002), had lower albumin levels (p = 0.014) and were more likely to receive cisplatin doses less than 40mg/m$^2$. There was a statistically insignificant trend

**Table 1. Female patient characteristics.**

| Characteristic | HIV-negative N = 40 | HIV-positive N = 32 | p-value |
|---|---|---|---|
| **Age**, n (%) | | | |
| <60 | 20 (50.0) | 27 (84.4) | **0.002** |
| 60+ | 20 (50.0) | 5 (15.6) | |
| **BMI**, n (%) | | | |
| <25 | 12 (70.0) | 16 (50.0) | 0.083 |
| ≥25 | 28 (30.0) | 16 (50.0) | |
| **BSA /m², median (IQR)** | 1.6 (1.5–1.8) | 1.62 (1.5–1.7) | 0.747 |
| **FIGO stage**, n (%) | | | |
| I | 2 (5.0) | 1 (3.1) | |
| II | 19 (47.5) | 19 (59.4) | 0.652 |
| III | 19 (47.5) | 12 (37.5) | |
| **Comorbidities**, n (%) | | | |
| None | 23 (57.5) | 25 (78.1) | |
| DM[a] and/ or HPT[b] | 17 (42.5) | 7 (21.9) | 0.065 |
| **Albumin**, median (IQR) | 40 (35–46.5) | 35 (33–40) | **0.014** |
| **Cisplatin BSA-dose mg/m²**, n (%) | | | |
| <40 | 25 (62.5) | 27 (84.4) | **0.040** |
| ≥40 | 15 (37.5) | 5 (15.6) | |
| **Number of treatment cycles**, median (IQR) | 3 (2–4.5) | 3 (2–4) | 0.467 |
| **Patients completing ≥4 cycles** | 17 (42.5) | 14 (23.8) | 0.915 |
| **Nominal dose mg**, median (IQR) | 60 (51.5–65) | 57.5 (50–61) | 0.386 |
| **Baseline eGFR ml/min**, median (IQR) | 80.5 (64–94.5) | 80.5 (65–91.5) | 0.814 |
| **cART[c] at diagnosis**, n (%) | | | |
| Yes | -- | 31 (96.9) | -- |
| No | | 1 (3.1) | |
| **Tenofovir containing cART**, n (%) | -- | 31 (96.9) | -- |
| **Time on cART years**, median (IQR) | -- | 5.5 (3–10) | -- |

[a] DM: diabetes mellitus

[b] HPT: hypertension

[c] Combination antiretroviral treatment

towards a higher prevalence of diabetes and/or hypertension in HIV-negative patients (42.5% vs 21.9%), p = 0.065. Patient characteristics are displayed in Table 1.

## Adverse renal events

The mean percent reduction in eGFR from baseline was -19% (95% CI: -25.9% to -13.2%) for all patients. Thirty-one (43.1%) patients experienced an adverse renal event (risk or injury) during the course of their treatment. When stratified according to HIV-status, the prevalence of renal impairment was 37.5% and 50.0% for HIV-negative and -positive patients respectively (p = 0.287). Other adverse renal events are displayed in Table 2. The differences between the two group were not statistically significant.

## Risk factors for renal impairment

Univariate odds for renal impairment (risk and/ or injury) were associated with age ≤ 60 years (OR 0.28, 95% CI: 0.09–0.82, p = 0.020), and baseline eGFR<60ml/min (OR 0.18, 95% CI:

**Table 2. Adverse renal events according to HIV status.**

| Event | HIV-negative N = 40 | HIV-positive N = 32 | p-value |
|---|---|---|---|
| **Lowest eGFR ml/min**, n (%) | | | |
| <60 | 11 (27.5) | 12 (37.5) | 0.365 |
| ≥60 | 29 (72.5) | 20 (62.5) | |
| **Change in eGFR**, mean (95% CI) | -14.9 (-21.2 to -8.6) | -17.9 (-26.7 to -7.6) | 0.682 |
| **RIFLE Class of impairment** n (%) | | | |
| None | 25 (62.5) | 16 (50.0) | 0.553 |
| Risk | 11 (27.5) | 11 (34.4) | |
| Injury | 4 (10.0) | 5 (15.6) | |

0.06–0.57, p = 0.003). Percent change of the lowest recorded eGFR from baseline are displayed in Fig 1. Lower reductions in eGFR from baseline were observed in patients with a baseline eGFR of <60ml/min and in those 60 years of age and older.

Potential confounding between age ≥ 60 years with eGFR<60 and dose administered was explored. Patients ≥60 years were 2.8 times (95% CI: 1.02–7.79) more likely to have a baseline eGFR<60ml/min (p = 0.041). Furthermore, the median cisplatin dose administered to patients with an eGFR<60ml/min was 50mg (IQR 54–68) vs 55mg (IQR 50–60) for those with eGFR≥60ml/min, Wilcoxon-rank test p = 0.044. This indicates that older patients were more likely to have renal insufficiency and be managed cautiously with lower doses.

Univariate and multivariate analysis of potential risk factors are presented in Table 3. HIV infection was not a predictor of renal impairment in adjusted and unadjusted analyses, adjusted OR 1.16 (CI: 0.35–3.43). Age ≥60 years, baseline eGFR≤60ml/min were associated with reduced odds of impairment on univariate analysis for all the patients. The only independent predictor of renal impairment was eGFR ≤60 ml/min with an OR 0.25 (95% CI: 0.07–0.85, p = 0.027).

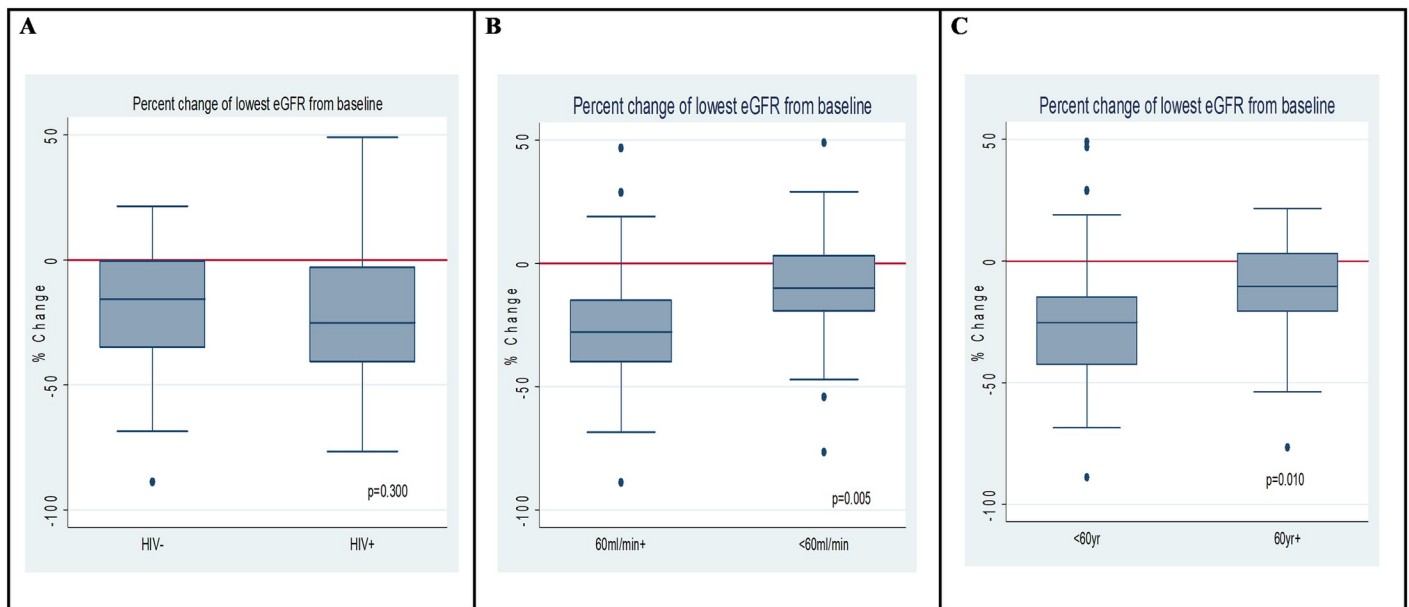

**Fig 1. Boxplot of percentage difference in baseline eGFR and lowest eGFR during cisplatin treatment according to A) HIV status B) Baseline eGFR < and ≥60ml/min C) Age < and ≥60 years of age.**

**Table 3. Univariate and multivariate logistic regression analysis of potential risk factors for renal impairment.**

| Risk factor | OR (95% CI) | P | Adjusted OR (95% CI) | P |
|---|---|---|---|---|
| HIV infection | 1.67 (0.65–4.28) | 0.289 | 1.16 (0.35–3.43) | 0.769 |
| Age ≥60 years | 0.23 (0.09–0.82) | **0.020** | 0.35 (0.10–1.19) | 0.093 |
| Stage III | 2.34 (0.90–6.10) | 0.082 | 1.96 (0.67–5.72) | 0.220 |
| BMI ≤25 | 1.25 (0.48–3.25) | 0.645 | -- | -- |
| Cisplatin cycles ≥4 | 1.46 (0.57–3.76) | 0.428 | -- | -- |
| Cisplatin dose ≥40mg/m$^2$ | 1.48 (0.52–4.16) | 0.462 | -- | -- |
| Cisplatin dose ≥ 50mg | 1.78 (0.62–5.14) | 0.288 | 1.44 (0.44–4.76) | 0.547 |
| Baseline eGFR <60ml/min | 0.18 (0.06–0.57) | **0.003** | 0.25 (0.07–0.85) | **0.027** |
| Albumin <35g/L | 1.99 (0.645–6.11) | 0.231 | -- | -- |

Within the HIV-positive group, duration of cART treatment was associated with an 8% increase in risk of renal impairment with each additional year of treatment (unadjusted OR 1.08, 95% CI 0.89–1.29, p = 0.434). This however did not reach statistical significance. Additionally, age ≥ 60 years was no longer significantly associated with risk of renal impairment OR 1.62, (p = 0.628, 95% CI 0.23–11.26). Unadjusted OR for eGFR within the HIV-positive stratum was 0.23 (p = 0.071, 95% CI 0.05–1.13).

## Discussion

HIV-infection not was associated elevated risk of renal impairment. There was a higher proportion of HIV-positive patients with renal impairment (50% vs 37.5%), this 12.5% difference was not statistically significant. These findings are consistent with previous observations where HIV-status did not influence non-haematological toxicities in patients receiving CCRT [19, 21]. All but one of the HIV-positive patients were receiving TDF-containing regimens, hence it was not possible to evaluate any additional risk posed by TDF. Within the HIV-positive population, there was an 8% increase in the risk of renal impairment for each additional year on cART however this was also statistically insignificant (p = 0.434).

The mean percentage reduction in eGFR from baseline was -19% (95% CI: -25.9% to -13.2%) for all patients, and the prevalence of renal impairment in the study population was 43.1%. The prevalence of renal impairment in this population was slightly higher than the 18–35% observed for cervical cancer patients in Thailand and Japan [9, 10]. In South Africa, the prevalence of renal toxicity in a similar group of patients receiving CCRT was only 17% [21]. In the South African study patients received a dose of 30mg/m$^2$, however higher doses administered (35-40mg/m$^2$) in this study may have contributed to the difference. The higher prevalence of renal impairment in this study may have been due to sub-optimal pre-hydration protocols. Race has been shown to increase the risk of renal impairment in patients with head and neck cancers treated with cisplatin [11], and may have contributed to the observations in this study. Alternatively, the high proportion of stage 3 disease may have contributed to the greater reductions in eGFR. Information on premedication with magnesium and potassium was not consistently available. The assumption that all patients received appropriate pretreatment was used, because it is recognized as standard of care. Notwithstanding, there is room for optimization of pre- and post-treatment hydration protocols used in the setting.

Patient characteristics between HIV-positive and–negative groups were generally similar, with the exception of age, albumin, cisplatin dose and BMI. Of the HIV-negative patients, 50% were <60 years of age vs 84.4% for HIV-positive patients (p = 0.02). It is generally accepted that HIV-positive patients with ICC tend to be younger than HIV-negative patients [22].

Unadjusted OR for albumin <35mg/dL was associated with a 99% increase in renal impairment, however this was not statistically significant p = 0.231(95% CI: 0.645–6.11). A larger sample size may be required to adequately assess the effects hypoalbuminaemia.

Dose adjustments of 25–50% are recommended for patient with an eGFR <60ml/min [23]. Baseline eGFR<60ml/min was independently associated with a 75% risk reduction in renal function (p = 0.03). Conflicting observations have been made with regard to the prognostic significance of renal insufficiency. Several studies found no association between baseline eGFR <60ml/min with renal impairment [9, 24, 25]. Similar to our findings, higher baseline creatinine clearance has been associated with increased risk of AKI in patients with head and neck cancers [11]. This may be due to receipt of lower cisplatin doses and more attention to hydration status and electrolyte levels. In this current study, there was a trend towards receipt of lower cisplatin doses in patients with low baseline eGFR (p = 0.065).

Study limitations were mainly due to its retrospective nature. Disease response and long-term renal outcomes were not captured in this study. This made it difficult to identify whether the cause of renal impairment was due to cisplatin or disease progression. Additionally, information on premedication was not consistently available, and it was assumed that all patients received standard of care. Numerically, HIV-infection led to an increase in renal impairment, however the sample size may have been too small to achieve statistical significance. The study assessed short-term renal outcomes and did not document renal recovery post-CCRT. Additional data on post-treatment renal outcomes would have demonstrated long term significance of study findings.

Patients in the current study were on non-nucleoside reverse transcriptase inhibitor containing regimens. As national HIV treatment programs guidelines transition toward dolutegravir- (DTG) containing regimens further studies may be required. Dolutegravir is a highly albumin bound drug which blocks renal organic cation transporter 2 (OCT2) which is partly responsible for cisplatin elimination [26]. As a result, a potential drug-drug interaction exists.

## Conclusions

HIV-infection was not associated with an increase in risk of renal impairment in ICC patients receiving definitive cisplatin-based concurrent chemoradiation (CCRT) in routine clinical care. The high prevalence of renal impairment in the study population suggests that pre-hydration protocols may need to be optimized.

## Supporting information

**S1 File. Data.**
(XLSX)

## Acknowledgments

We would like to thank Dr Mazhindu who was very valuable during data collection.

## Author Contributions

**Conceptualization:** Pinky M. C. Manyau, Mensil Mabeka, Webster Kadzatsa.

**Data curation:** Mensil Mabeka, Webster Kadzatsa.

**Formal analysis:** Pinky M. C. Manyau, Mensil Mabeka, Tinashe Mudzviti, Albert Nyamhunga.

**Funding acquisition:** Mensil Mabeka.

**Methodology:** Pinky M. C. Manyau, Tinashe Mudzviti, Webster Kadzatsa.

**Project administration:** Pinky M. C. Manyau.

**Resources:** Mensil Mabeka.

**Supervision:** Pinky M. C. Manyau, Webster Kadzatsa.

**Writing – original draft:** Pinky M. C. Manyau, Tinashe Mudzviti, Webster Kadzatsa, Albert Nyamhunga.

**Writing – review & editing:** Pinky M. C. Manyau, Tinashe Mudzviti, Webster Kadzatsa, Albert Nyamhunga.

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
