## [Decision Letter · Decision Letter 0]

20 Nov 2020

PONE-D-20-14840

Renal function impairment in cervical cancer patients treated with cisplatin-based chemoradiation:

PLOS ONE

Dear Dr. Manyau,

Thank you for submitting your manuscript to PLOS ONE. After careful consideration, we feel that it has merit but does not fully meet PLOS ONE’s publication criteria as it currently stands. Therefore, we invite you to submit a revised version of the manuscript that addresses the points raised during the review process.

We look forward to receiving your revised manuscript.

Kind regards,

Peng Gao, Ph.D. & M.D.

Academic Editor

PLOS ONE

Journal Requirements:

2. In the ethics statement in the manuscript and in the online submission form, please provide additional information about the patient records used in your retrospective study. Specifically, please ensure that you have discussed whether all data were fully anonymized before you accessed them and/or whether the IRB or ethics committee waived the requirement for informed consent. If patients provided informed written consent to have data from their medical records used in research, please include this information."

3. To comply with PLOS ONE submission guidelines, in your Methods section, please provide additional information regarding your statistical analyses. For more information on PLOS ONE's expectations for statistical reporting, please see https://journals.plos.org/plosone/s/submission-guidelines.#loc-statistical-reporting.”

4. Please include a caption for figure 1.

Reviewers' comments:

Reviewer's Responses to Questions

**Comments to the Author**

1. Is the manuscript technically sound, and do the data support the conclusions?

Reviewer #1: Yes

2. Has the statistical analysis been performed appropriately and rigorously? 

Reviewer #1: Yes

3. Have the authors made all data underlying the findings in their manuscript fully available?

Reviewer #1: Yes

4. Is the manuscript presented in an intelligible fashion and written in standard English?

Reviewer #1: Yes

5. Review Comments to the Author

Reviewer #1: This paper studied the relationship between HIV-infection and increased renal impairment in ICC patients in Zimbabwe. They found the prevalence of renal impairment in the study is higher than patients in other counties and gave some suggestions for the treatment.This study is well-designed and the results are reliable. However, there are some issues exist.

1. The first paragraph needs a subtitle.

2. The subtitle "Changes in renal function" should be changed to a more suitable one to reflect the findings you described.

3. Both "GFR" and "eGFR" are used in this manuscript. What is the difference between them?

4. All 3 panels in Figure 1 lack the names y-axis. Also, it is better to add P values to all 3 panels.

6. PLOS authors have the option to publish the peer review history of their article (what does this mean?). If published, this will include your full peer review and any attached files.

Reviewer #1: No

---

## [Author Response · Author response to Decision Letter 0]

22 Dec 2020

The manuscript has been revised as follows:

- Line numbers have been inserted;

- Level 1 and 2 headings have been inserted for major and minor headings respectively;

- “Figure 1” has been converted to .tif format.

2. In the ethics statement in the manuscript and in the online submission form, please provide additional information about the patient records used in your retrospective study. Specifically, please ensure that you have discussed whether all data were fully anonymized before you accessed them and/or whether the IRB or ethics committee waived the requirement for informed consent. If patients provided informed written consent to have data from their medical records used in research, please include this information."

The ethics statement has been revised to:

“All data were fully anonymized before they were accessed. Ethics approvals were obtained from the Joint Research and Ethics Committee for University of Zimbabwe College of Health Sciences and Parirenyatwa Group of Hospitals (JREC#:315/19) before commencement of the study”.

3. To comply with PLOS ONE submission guidelines, in your Methods section, please provide additional information regarding your statistical analyses. For more information on PLOS ONE's expectations for statistical reporting, please see https://journals.plos.org/plosone/s/submission-guidelines.#loc-statistical-reporting.”

More detail has been provided for the following:

- Descriptive statistics;

- Baseline comparisons between HIV-infected and -uninfected patients;

- All p-values have been reported to 3 decimal places;

- Use and presentation of odds ratios obtained from regression analysis

Table 3 which presents results for regression analysis has also been amended. The 95% confidence interval which was initially in the same column as the p-value has been placed in the same column as the point estimate.

4. Please include a caption for figure 1.

The caption for figure 1 has been inserted after the paragraph where the figure is first cited (line 161-163).

The heading “Supporting Information” along with the description (“Data”)

Reviewer #1: This paper studied the relationship between HIV-infection and increased renal impairment in ICC patients in Zimbabwe. They found the prevalence of renal impairment in the study is higher than patients in other counties and gave some suggestions for the treatment. This study is well-designed and the results are reliable. However, there are some issues exist.

1. The first paragraph needs a subtitle.

The first paragraph has been titled “Background”

2. The subtitle "Changes in renal function" should be changed to a more suitable one to reflect the findings you described.

The section reports the distribution of adverse renal events (lowest eGFR <60ml/min, percent reductions in eGFR, and grading of adverse renal events), hence the subtitle has been changed to “Adverse renal events”.

3. Both "GFR" and "eGFR" are used in this manuscript. What is the difference between them?

We reported estimated GFR based on creatinine clearance hence the term eGFR has been adopted for consistent use.

4. All 3 panels in Figure 1 lack the names y-axis. Also, it is better to add P values to all 3 panels.

The y-axis was labelled “% Change”, and the respective p=values have been embedded in the figure.

---

## [Editor Report · Decision Letter 1]

30 Dec 2020

Renal function impairment in cervical cancer patients treated with cisplatin-based chemoradiation: A review of medical records in a Zimbabwean outpatient department

PONE-D-20-14840R1

Dear Dr. Maudy,

We’re pleased to inform you that your manuscript has been judged scientifically suitable for publication and will be formally accepted for publication once it meets all outstanding technical requirements.

Kind regards,

Peng Gao, Ph.D. & M.D.

Academic Editor

PLOS ONE

---

## [Editor Report · Acceptance letter]

5 Jan 2021

PONE-D-20-14840R1 

Renal function impairment in cervical cancer patients treated with cisplatin-based chemoradiation: A review of medical records in a Zimbabwean outpatient department 

Dear Dr. Manyau:

I'm pleased to inform you that your manuscript has been deemed suitable for publication in PLOS ONE. Congratulations! Your manuscript is now with our production department. 

Kind regards, 

on behalf of

Dr. Peng Gao 

Academic Editor

PLOS ONE